# Diffracted X-ray Tracking for Observing the Internal Motions of Individual Protein Molecules and Its Extended Methodologies

**DOI:** 10.3390/ijms241914829

**Published:** 2023-10-02

**Authors:** Yuji C. Sasaki

**Affiliations:** 1Graduate School of Frontier Sciences, The University of Tokyo, 5-1-5 Kashiwanoha, Chiba 277-8561, Japan; ycsasaki@edu.k.u-tokyo.ac.jp; 2AIST-UTokyo Advanced Operando-Measurement Technology Open Innovation Laboratory (OPERANDO-OIL), National Institute of Advanced Industrial Science and Technology (AIST), 6-2-3 Kashiwanoha, Chiba 277-0882, Japan; 3Center for Synchrotron Radiation Research, Japan Synchrotron Radiation Research Institute, 1-1-1 Kouto, Sayo-cho 679-5198, Japan

**Keywords:** diffracted X-ray tracking, conformational protein dynamics, single molecule observations using X-rays, diffracted X-ray blinking

## Abstract

In 1998, the diffracted X-ray tracking (DXT) method pioneered the attainment of molecular dynamics measurements within individual molecules. This breakthrough revolutionized the field by enabling unprecedented insights into the complex workings of molecular systems. Similar to the single-molecule fluorescence labeling technique used in the visible range, DXT uses a labeling method and a pink beam to closely track the diffraction pattern emitted from the labeled gold nanocrystals. Moreover, by utilizing X-rays with extremely short wavelengths, DXT has achieved unparalleled accuracy and sensitivity, exceeding initial expectations. As a result, this remarkable advance has facilitated the search for internal dynamics within many protein molecules. DXT has recently achieved remarkable success in elucidating the internal dynamics of membrane proteins in living cell membranes. This breakthrough has not only expanded our knowledge of these important biomolecules but also has immense potential to advance our understanding of cellular processes in their native environment.

## 1. History and Concept of Single Molecule Dynamics Using X-rays

Single-molecule measurements using visible light, which started in the 1970s, have greatly advanced our understanding of the translational movement of single protein molecules moving in living cells and their real-time single-molecule interactions with other protein molecules [1,2]. In addition, the fluorescence resonance energy transfer (FRET) method using visible light was partially successful in monitoring the intramolecular motions of proteins [3]. However, this FRET method did not develop into a general-purpose method for measuring changes in the internal conformation of molecules. To measure the internal motions of protein molecules with high precision, it was necessary to observe the dynamics at atomic wavelengths; thus, expectations for X-rays, electron beams, and neutrons naturally increased [4,5,6,7,8,9]. 

Once the methodology for determining protein molecular structures using X-rays and electron beams was established, the next necessary information for understanding the mechanisms of protein molecular functions in more detail was the molecular conformational changes with atomic-level precision that accompany the functional expression of protein molecules. The actual observation of the internal motion of a single protein molecule started with the interpretation of multiple molecular structures attributed to single-particle analysis by cryo-electron microscopy [10,11,12]. However, this observation of internal molecular motion was a prediction and only a dynamic prediction from the static images of multiple molecular structures. The direction and speed of their protein motion, as well as their stability, cannot be rigorously determined. Continuous molecular structure and dynamic information need to be acquired. However, the quantum beam probes at this time were not sensitive enough to detect a single molecule. 

A methodology for measuring the internal dynamics of single molecules using X-rays proposed in 1998 was the diffracted X-ray tracking (DXT) method; in DXT, gold nanocrystals were chemically labeled at sites of structural change, and diffraction spots from these nanocrystals were monitored over time [13,14,15,16,17,18,19,20]. This method was called the X-ray single-molecule tracking method. As shown in Figure 1, the principle is quite simple: one diffraction spot from one labeled nanocrystal is tracked on time-resolved diffraction images to determine whether the labeled nanocrystal and the labeled site of the protein molecule are in the same motion. DXT measurements on various systems have shown that in the microsecond to millisecond range, the real-space rotational motion of the nanocrystal diffraction spots coincides with the rotational motion of the domain containing the protein labeling sites. In some cases, this can also be determined from the correlation analysis between the size of the labeled nanocrystal and the motion of the diffraction spots, which are slightly slower due to the motion suppression effect of the nanocrystal labeling [20,21,22,23,24,25]. 

The main dynamic information of the DXT is found on the two axes of the 2D diffraction image. These are the θ and χ directions shown in Figure 2. Regarding the two axes on the diffraction image, the region where diffraction points occur is limited by the X-ray energy width. However, because of this limited space, the motion of the diffraction points can be observed in the same direction as the internal motion of the molecule, as in real space. In particular, the rotational motion of protein molecules is directly related to their function. For example, considering that membrane proteins are usually formed in membrane polymers, the opening and closing of the channels, which are the most important functions of the membrane proteins, is not a translational movement but is achieved by rotation and twisting of each domain. The ability to directly confirm this rotation by zooming in on the X-ray diffraction image at high speed is extremely important information for the DXT method. Another advantage is that the rotational motion is reversed when the sample substrate is changed from upstream to downstream, as shown in Figure 3; this process confirms the direction of rotation, causing an easy check for reproducibility. Although tilting and twisting models are sometimes proposed for the internal dynamics of membrane protein molecules, as shown in Figure 4, the greatest advantage of DXT is that these two models can be easily distinguished as a combination of the θ and χ motions of DXT [26,27,28,29]. 

The resolution of θ and χ, which are crucial for DXT measurements, is determined by the pixel size of the two-dimensional detector and the distance between the camera and the sample, known as the camera length. For instance [26], in our DXT measurements, we recorded time-resolved diffraction images using an X-ray image intensifier (V5445P, Hamamatsu Photonics) and a CMOS camera (1024 pixels × 1024 pixels, Photron FASTCAM SA1.1). This FASTCAM SA1.1 high-speed camera offers exceptional speed, capturing up to 9000 frames per second, and delivers true 12-bit performance (dynamic range). The nominal entrance field of view for the X-ray image intensifier is 150 mm in diameter, with an effective pixel size of 0.1465 mm. With the incident X-ray’s peak energy set at 15.2 keV and a sample-to-detector distance of 100 mm in our DXT setup, a one-pixel movement of a diffraction spot in the tilting (θ) direction corresponds to 0.7 mrad/pixel (@15.2 keV). For most diffraction spots originating from gold nanocrystals situated 36.4 mm from the beam center, considering the d-spacing of Au (111) (d = 2.35 Å), this distance corresponds to 248.5 pixels in our configuration. The circle with a radius of 248.5 pixels corresponds to approximately 1560 pixels in circumference. Consequently, a one-pixel shift in the twisting (χ) direction corresponds to 4.0 mrad/pixel @15.2 keV. 

The main dynamic information in the DXT is the two axes of the 2D diffraction image. These are the θ and χ directions shown in Figure 2. The environment around the sample is shown in Figure 5. Currently, DXT measurements are obtained in various places, as shown in Figure 6; for the two axes on the diffraction image obtained by DXT, as shown in Figure 2, the region where the diffraction points occur is limited by the X-ray energy width. However, because of this limited space, the motion of the diffraction points can be observed in the same direction as the internal motion of the molecule, as in real space. 

## 2. Nanocrystal Labeling for DXT

DXT requires a nanocrystal labeled for a target biomolecule. The nanocrystals to be labeled need to be as compact as possible and able to efficiently label protein molecules; early experiments with DXT have mostly targeted and chemically modified SH groups introduced into DNA and Cys and Met in protein molecules. The reaction of gold with Cys and Met has been shown to form very efficient and robust reactions in studies on self-assembled membranes. Gold nanocrystals are stably sold as gold colloids as fine particles of various sizes. However, when the gold colloids are, for example, 40–60 nm in diameter, it is not possible to trace the diffraction points; this is the basis of DXT since the X-ray diffraction rings can be observed but not as diffraction spots [30]. Depending on the manufacturer, diffraction spots can only be observed when the gold colloid diameter is approximately 100 nm. Gold colloids do not effectively crystallize, as many gold colloids have been adapted to low-temperature production methods.

From the beginning, we have used epitaxial growth on single crystals of sodium chloride and potassium chloride to produce gold nanocrystals with good crystallinity and small diameters, using single crystal fabrication techniques at high vacuum temperatures of 400–500 °C [30,31]. The diffraction spots from the lab-made gold nanocrystals are shown in Figure 7 [30]. The size of the gold nanocrystals in the single crystals produced can be freely controlled by the substrate temperature, deposition rate, and deposition amount. To date, sizes of 20–200 nm in diameter have been fabricated.

The next important point is that the gold nanocrystals fabricated on the single-crystal substrate in the vacuum evaporation system need to be uniformly dispersed in the aqueous solution, as shown in Figure 8. The preparation of this aqueous solution of well-dispersed gold nanocrystals is very important because the protein molecules react with the gold nanocrystals in the aqueous solution. If there are no restrictions on the protein molecules before the reaction, the substrate exposed to an aqueous solution of 0.1–1 mM surfactant immediately disperses in it; thus, the substrate is agitated and stabilized to some extent via an ordinary ultrasonic cleaner. Aqueous gold nanocrystal solutions with added surfactants can usually maintain their dispersibility stably for 1–2 h. However, surfactants cannot be used when the substrate is immobilized for measurements of membrane protein molecules or cells.

The surface of the deposited, fabricated gold nanocrystals is water-repellent. For example, if the self-assembled film can react with the water-repellent surface of that gold nanocrystal in an ethanol solution, the gold nanocrystal surface can be made hydrophilic, as shown in Figure 8b [30]. In fact, anhydrous ethanol does not dissolve potassium chloride; thus, it is easy to perform the above reaction, which takes 1–2 h. However, only half of the gold nanocrystal surface can be hydrophilized; notably, we have confirmed that gold nanocrystals can be stably solubilized for 1–2 h with that level of hydrophilization. Using the same concept, we also confirmed that after the gold nanocrystals are fabricated, the dispersion state is maintained to some extent even when stable oxides, such as aluminum or titanium dioxide, are vacuum deposited, as shown in Figure 8b [30]. 

The most stable way to produce a dispersible aqueous solution of gold nanocrystals is as follows: Gold nanocrystals can be made very stable when reacted in aqueous solution with an antibody against that protein molecule. Additionally, the specific reactivity of the antibody with the protein, which was a concern, was maintained. However, since the antibody molecule is a comparatively large molecule, there was a possibility that the observation of the protein binding site and the movement of the binding domain, which was the original aim of the observation, could potentially be slowed down. In these cases, the use of special antibodies, such as Fab antibodies or microantibodies, could be considered. The use of antibodies is very effective in terms of specific labeling of protein molecules. However, care needs to be taken with some proteins since the original function of the protein molecule can potentially be impaired when they are reacted with antibodies. 

As shown in Figure 9, DXT is applicable to a variety of proteins [32,33,34,35,36,37,38]. From these many DXT measurements, labeling larger gold nanocrystals slightly reduces the measured kinetic size, as shown in Figure 10 [26].

## 3. Substrate Immobilization Method for DXT

In DXT, the protein of interest is fixed to the substrate [39,40,41]. Recently, living cells were absorbed in substrates, and the membrane proteins present in the cell membrane were observed [42,43,44]. The method of substrate immobilization of protein molecules is similar to the method of labeling protein molecules with gold nanocrystals. For substrate fixation of the protein molecules, if the protein molecules have Cys or Met, the protein molecules are reaction-fixed by simply depositing an evaporated gold thin film on a polyimide film with a thickness of 5–20 microns; this film is resistant to X-ray durability and does not peel off. However, if the direction of rotation of motion is to be experimentally established using DXT, a molecular orientation is needed to provide meaning to the analysis of the direction of rotation of the molecule, as shown in Figure 11, where the vertical orientation of the protein molecule can be determined. A method often used is the introduction of Tag, which is used to create protein mutants. If the substrate side can be modified with a surface modification that reacts easily with a His-tag, the orientation of the protein is easily controlled by binding the His-tag-introducing side to the substrate side. Other Fc-tag and various tag orientation techniques are available, all of which can be used. When the cells are attached to the substrate, the substrate can be attached through charges with a normal poly-L-lysine coat.

## 4. Analysis Methods for Diffraction Point Tracking Data

The basic analysis concept of single-molecule measurements using visible light has not been used to discuss a single event. This needs to be strictly followed in DXT using X-rays; single molecule data needs to be collected and statistically processed; the basic observed phenomenon in DXT is the Brownian motion of the site of interest of the protein molecule. This random motion greatly differs under different physiological conditions of the protein and in the presence of ligands. Therefore, all conventional methods for analyzing Brownian motion can be used. The most representative is the MSD curve. This analysis method is not a single molecule measurement result. It is an important guideline as a first deduction as to whether it is in line with normal Brownian motion. Normally, the θ and χ directions are independently evaluated, as shown in Figure 12 [26]; however, in some cases, the sum of the motions in the θ and χ directions is evaluated, as the θ and χ directions often simultaneously move.

Since the MSD curve is an average data analysis, this rational strategy can be used to analyze the motion distribution from single-molecule measurements. For example, the determination of the numerous different modes of motion can be analyzed in each θ and χ direction by fitting with a one-dimensional Gaussian distribution representation, as shown in Figure 13 [21]. Alternatively, this one-dimension Gaussian distribution can be represented in two dimensions, as shown in Figure 14 [26]. In this case, 2D Gaussian fitting can be used to analyze complex motion modes with closely related θ-χ. An example of using this 2D histogram notation for functional analysis is shown in Figure 15 [26]. The difference between the two-dimensional histograms when the ligand is not bound and when it is bound can be differentiated to clarify the difference in motion of the two axes (θ and χ) in the presence and absence of the ligand. Notably, although θ and χ are not real spaces, they can be interpreted as 3D structure change notations as they can be transformed into polar coordinates. If these data are created as a movie, a four-dimensional structure change profile can be represented as a two-dimensional movie. The final DXT movie provides an easy-to-understand notation for functional analysis based on the internal dynamics of protein molecules. 

Here, the validity of many sequential data points from time-resolved measurements, such as DXT, needs to be explained. DXT uses X-rays, which can deform protein molecules by imparting large amounts of energy to them. If this can be quantitatively assessed, the internal dynamics of the molecule can be reliably measured. For example, this problem can be solved by independently analyzing a series of 2D images in the first half and the second half, as shown in Figure 16 [44], to obtain an MSD curve, as shown in Figure 12. If there is molecular deformation or damage caused by X-rays, the effects should be cumulative over time. This can be solved by comparing the results of the first half of the series of images with those of the second half.

Another advantage of DXT is its motion accuracy. A simple motion measurement principle is shown in Figure 17. The measurement probe uses X-rays with a short wavelength, enabling the measurement of structural changes with very high accuracy. Notably, we are not measuring the translational displacement distance as in electron microscopy or atomic force microscopy (AFM), but the angle of change as shown in Figure 17 [13]. For example, if we monitor a 100 mrad angular change in a molecule of 10 nm length, the translational change would be a small translational distance of 1 nm. In X-ray diffraction measurements, the diffraction angle can be determined to an accuracy of 0.1 mrad, depending on the camera length. If there is a change of 1 mrad, this can be converted into a translational travel distance of 1 picometer, as shown in Figure 17 [13]. This measurement accuracy is remarkable.

Based on the above data, it is possible to immediately understand which movements are specifically expressed; for example, the presence and absence of the ligand can be obtained by taking the difference between the two-dimensional movement histograms for each condition. As shown in Figure 16, the ability to precisely adapt the evaluation of a motion to the measurement of the internal dynamics of molecules that play a highly important role in biological phenomena, such as membrane protein molecules, is a major feature.

## 5. Overcoming the Peculiarities of DXT

DXT is the only X-ray measurement technique that can precisely measure the 3D conformational change information of protein molecules. However, it is a rather specialized measurement technique and uses synchrotron white light. The essential principle of DXT is to track diffracted X-rays, and white X-rays with a certain energy range are needed. If monochromatic X-rays were used to monitor the motion of diffraction points on gold nanocrystals labeled with protein molecules, the diffraction point would disappear as soon as the diffraction point moved. Since the motion is not in one direction but in random motion, the observation of the diffraction point in motion using monochromatic X-rays, which have only a narrow energy range, causes the appearance of a blinking diffraction point. This concept is the principle of the diffracted X-ray blinking method DXB, as shown in Figure 18 [45,46,47,48,49,50,51,52,53,54]. Unlike DXT, where the exact direction of motion can be determined, the speed of motion can be monitored; the X-ray blinking phenomenon is similar to the X-ray Photon Correlation Spectroscopy: XPCS analysis method [55,56], which is an X-ray technique. Using the autocorrelation function, the speed of motion can be determined from the blinking phenomenon. Although this kinetic information does not reach the DXT information, it is very useful for assessing the internal dynamics of protein molecules. In addition, DXB using monochromatic X-rays can be used with laboratory X-ray sources without using synchrotron radiation. 

Currently, only a limited number of synchrotron radiation beamlines have white X-rays available. Therefore, fast DXB using monochromatic X-rays from synchrotron radiation is highly needed; the major analytical difference between DXT and DXB is that DXT only tracks the motion positions of diffraction points for the statistically necessary 100–10,000 points. However, DXB requires precise monitoring of the intensity changes of the diffracted X-rays, as shown in Figure 19 [42]. Currently, many beamlines, such as synchrotron radiation, are intense, but the X-ray intensity is time-dependent and unstable; DXB analysis needs a more complex quantitative analysis of the X-ray intensity than DXT analysis. 

Another disadvantage of DXT is its use of a labeling method, i.e., labeling gold nanocrystals. The advantage of single-molecule measurements in relatively complex systems, such as cell surfaces and cell interiors, is simple and good if they can be chemically labeled precisely at a particular location. However, the kinetic modulation of the gold nanocrystals due to labeling and the uncertainty as to whether they are precisely labeled at the target position remain problems. Fortunately, given the absence of any observed unidirectional increase in the motion of gold nanocrystals in numerous DXT measurements conducted thus far, we posit that the consideration of thermal excitation of gold nanocrystals during synchrotron radiation is unnecessary. 

Our next aim is to enable single-molecule dynamic measurements without labeling gold nanocrystals since direct detection of protein molecules is not possible with XFEL due to its sensitivity. Therefore, we could potentially modify the concept and just measure the information on the internal motion of the molecule, not a single molecule. Although a size problem with the internal motion of molecules is present, these measurements may be possible in the small-angle scattering region rather than the wide-angle scattering region. Small-angle scattering measurements on proteins are commonly known as SAXS, and the presence or absence of intramolecular motion by measuring SAXS in a time-resolved manner and obtaining an autocorrelation function on each pixel can be determined. 

As described above, a measurement technique that avoids the drawbacks of DXT (the use of white X-rays and gold nanocrystals) can also be proposed. Methodologically, the possibilities of DXT for the analysis of the wide-angle X-ray scattering regions using white X-rays, DXB for the analysis of the wide-angle X-ray scattering regions using monochromatic X-rays, and SAXB (Small-angle X-ray Blinking) for low-angle X-ray scattering regions using monochromatic X-rays could be discussed. As an extension of these, the possibility of dynamic measurements of large movements of protein molecules in the transmission X-ray region needs to be potentially monitored. At that time, the method will be called TXB (Transmitted X-ray Blinking).

In the future, in vivo DXT/DXB will likely have an even greater contribution to the effective elucidation of biological phenomena. Many technological advances can also be expected [53,54,57,58,59]. For example, this technology is expected to be adapted to electron and neutron beams instead of X-rays. In principle, the electron-beam single molecule tracking method has been successful [60,61,62,63,64]; in DET, a special sample holder was used, as shown in Figure 20a. Additionally, in principle, DXT with monochromatic X-rays can also be attained by the fabrication of special nanoparticles, as shown in Figure 20b. 

## Figures and Tables

**Figure 1 ijms-24-14829-f001:**
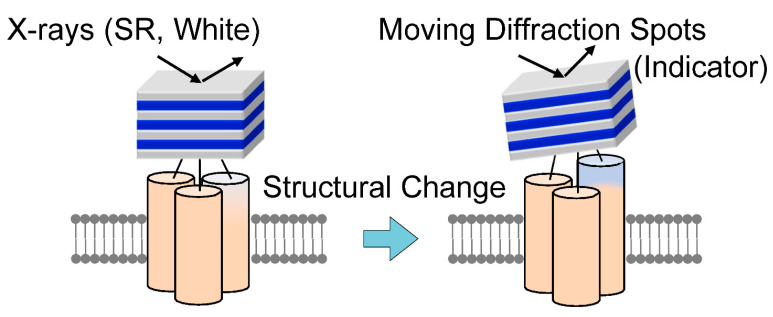
A principle diagram of single-molecule protein dynamic measurements using X-rays. A gold nanocrystal is labeled at the motion site of interest, and the positions of the X-ray diffraction spots diffracted by the labeled gold nanocrystal are tracked in a time-resolved manner. To track X-ray diffraction spots, the X-rays need to have a certain wavelength width.

**Figure 2 ijms-24-14829-f002:**
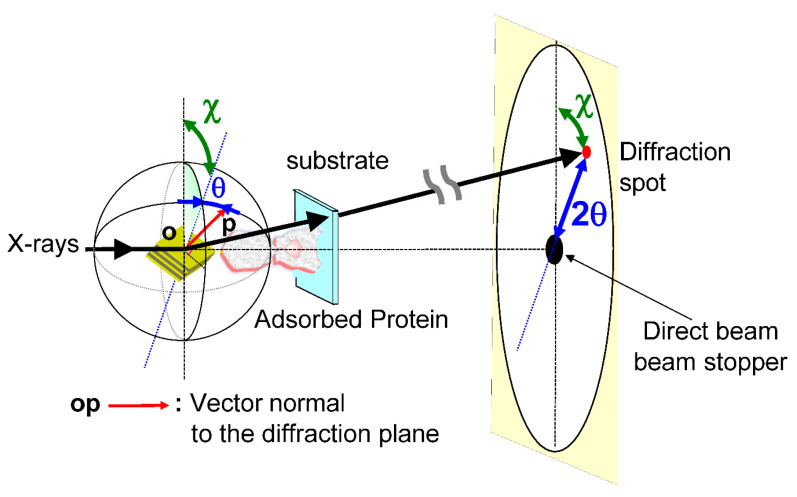
Schematic arrangement of the DXT instrumentation using the Laue method instrumentation for tracking X-ray diffraction. When tracking X-ray diffraction spots in time, the position of θ and χ at each measurement time is important.

**Figure 3 ijms-24-14829-f003:**
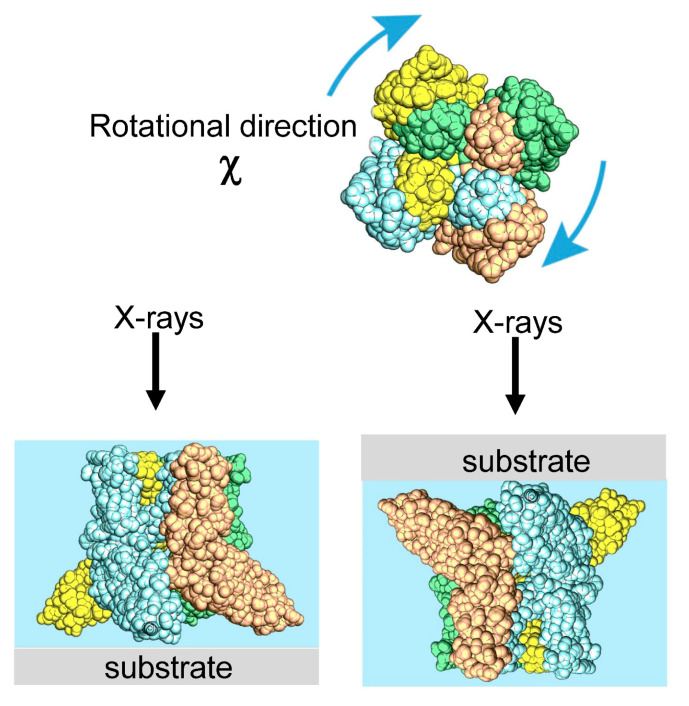
Various methods exist for the adsorption of protein molecules onto DXT substrates. Since the molecules can be adsorbed to the substrate in an aligned orientation, the rotational motion (χ direction) of the molecules can be rotated in the desired direction, and the direction of rotation can be reversed for observation. This can also be effectively used for data evaluation. When the surface-adsorbed monolayer protein molecule is a two-dimensional crystalline film, the DXT data would pertain to the θ direction rather than the χ direction.

**Figure 4 ijms-24-14829-f004:**
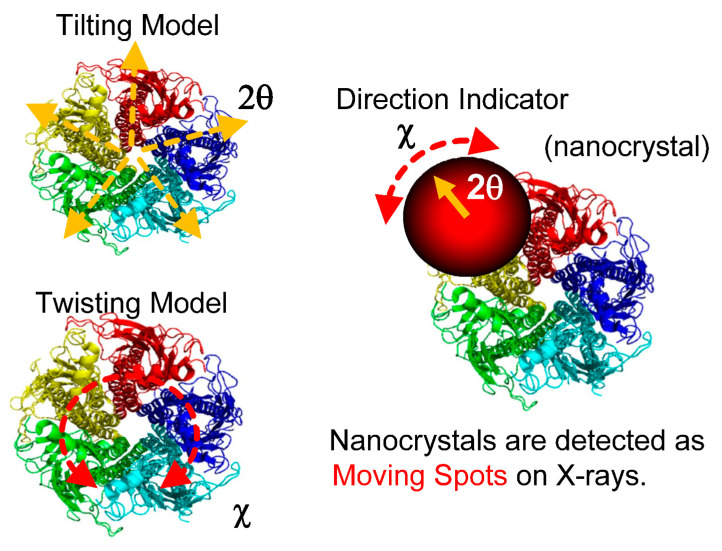
Channel molecules have two conformational dynamics; channel molecules are the most important and representative of membrane protein molecules. DXT can easily determine which of the two (the tilting model or the twisting model) is correct. In the tilting model, the motion in the θ direction significantly changes with the opening and closing motions of the channel. In the Twisting model, the motion in the θ direction in the opening and closing motions is slight; however, the motion in the χ direction is large.

**Figure 5 ijms-24-14829-f005:**
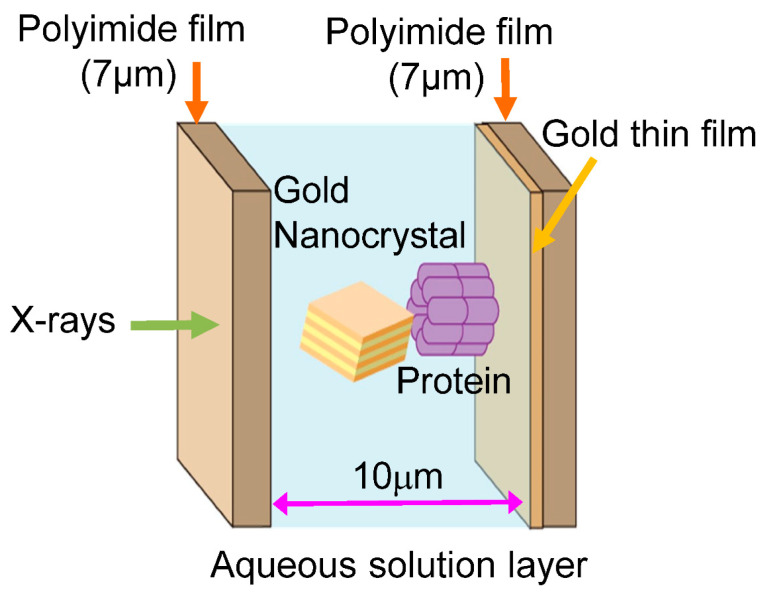
Cross-section of a DXT sample. A protein molecule is adsorbed and immobilized on one side, and each protein molecule is chemically labeled with one gold nanocrystal. The aqueous solution layer is sandwiched between polyimide films and made as thin as 10 microns to minimize X-ray scattering from the aqueous solution layer. The polyimide film surface can be deposited with gold or chemically treated to react with protein molecules.

**Figure 6 ijms-24-14829-f006:**
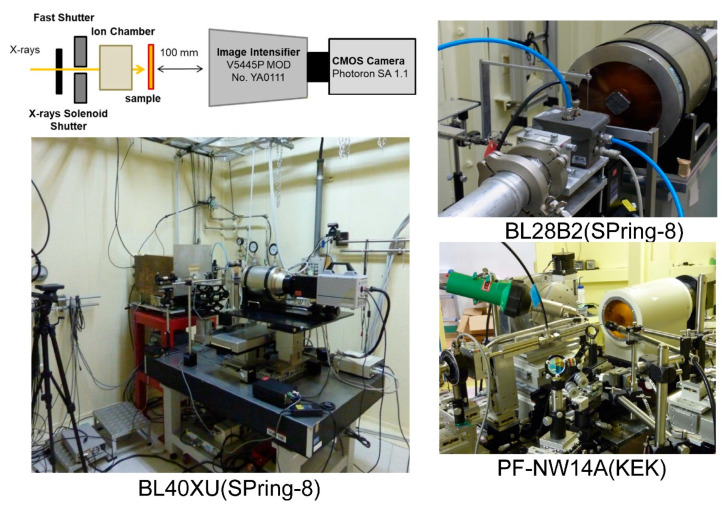
There are three main beamlines for DXT in Japan, with BL40XU at SPring-8 being the most suitable DXT beamline in terms of X-ray intensity and X-ray energy width (approximately 4%). Typically, time-resolved DXT of 10–100 microseconds can be performed. Recently, it has become possible to measure DXT while performing laser excitation.

**Figure 7 ijms-24-14829-f007:**
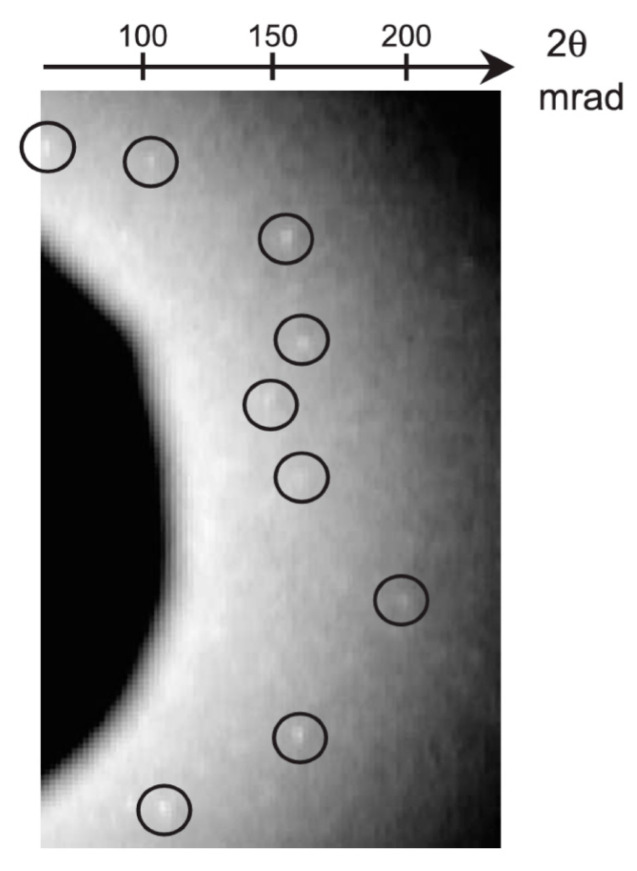
Gold nanocrystals are used in DXT [30]. Au nanocrystals with good crystallinity that can be produced using a unique fabrication method are ideal for DXT, as diffraction spots were surrounded by a circle can be measured from each individual crystal.

**Figure 8 ijms-24-14829-f008:**
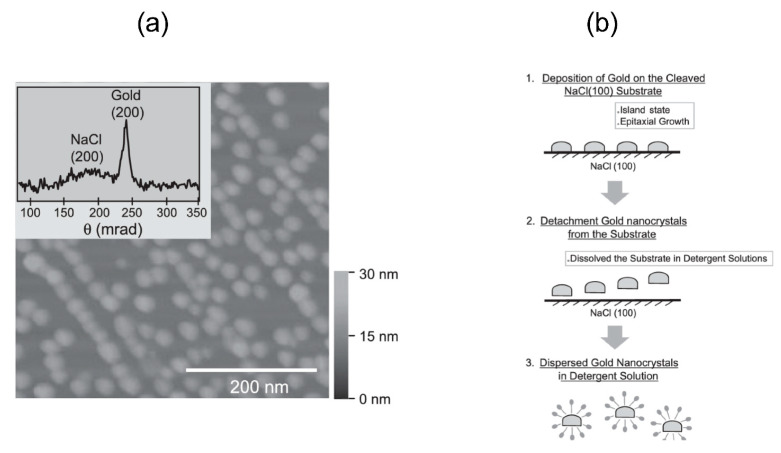
(**a**) AFM image of a gold nanocrystal on a KCl (100) single crystal of a lab-made gold nanocrystal [30]. The diameter of the gold nanocrystals is approximately 20–40 nm. Inserted in the box are X-ray diffraction patterns from the island-shaped gold particles on the NaCl substrate. (**b**) Hydrophilic gold nanocrystal fabrication process: chemical modification and oxide deposition on the surface of gold nanocrystals to ensure good dispersion of the gold nanocrystals produced in the lab [30]. Modification of only half the surface clearly changes the dispersibility of gold nanocrystals in an aqueous solution. For example, extremely high-quality, dispersive gold nanocrystal solutions can be produced by surface-reacting antibodies with gold nanocrystals.

**Figure 9 ijms-24-14829-f009:**
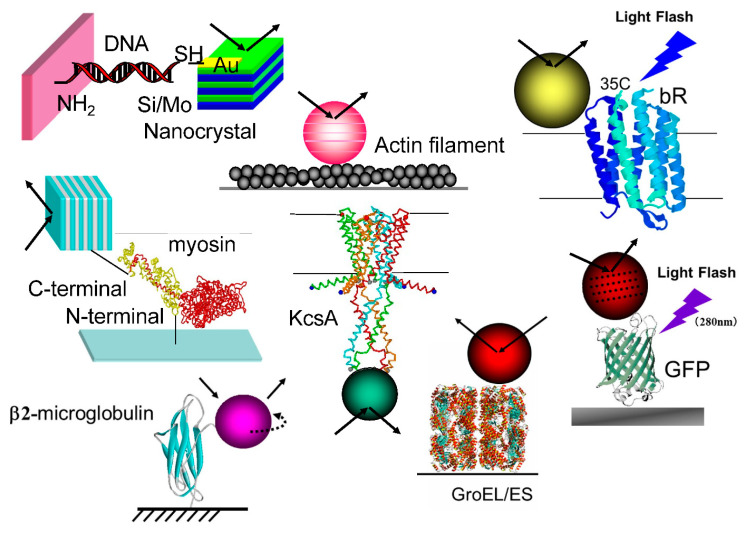
Measurements of the single-molecule dynamics of many proteins—single molecules from DNA to protein molecules—have been successfully measured.

**Figure 10 ijms-24-14829-f010:**
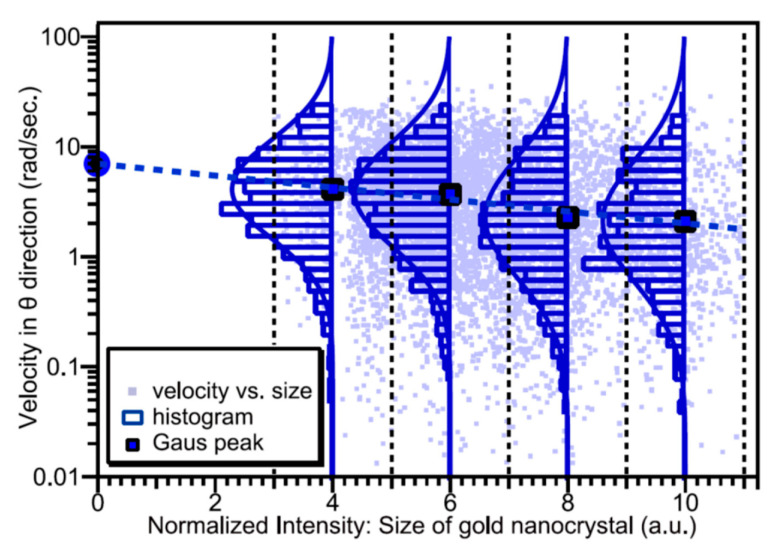
Detection motion size of proteins as a function of the size of the labeled gold nanocrystals [26]. As the size of the gold nanocrystal increases, the motion speed of the observed protein molecule slows down but does not significantly change. By scanning the size of the gold nanocrystal, the size of the intramolecular motion of non-labeled proteins can also be determined.

**Figure 11 ijms-24-14829-f011:**
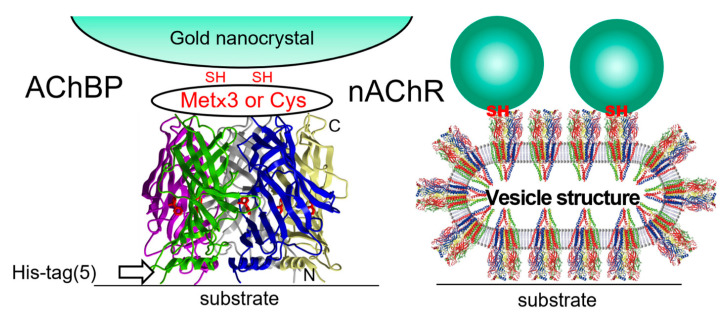
DXT measurements of single membrane protein molecules in the plasma membrane of living cells. However, the distance to neighboring membrane protein molecules cannot be controlled; thus, two or more membrane protein molecules may bind to a single gold nanocrystal. Two examples of protein molecules are shown: AChBP is a pentamer with a histag at the N-terminus, which is attached to the substrate. The reaction with gold nanocrystals is chemically modified with Cys and Met. The membrane protein nAChR is chemically labeled with Cys and gold nanocrystals of a nonsubstrate-adsorbed membrane protein molecule for DXT measurements.

**Figure 12 ijms-24-14829-f012:**
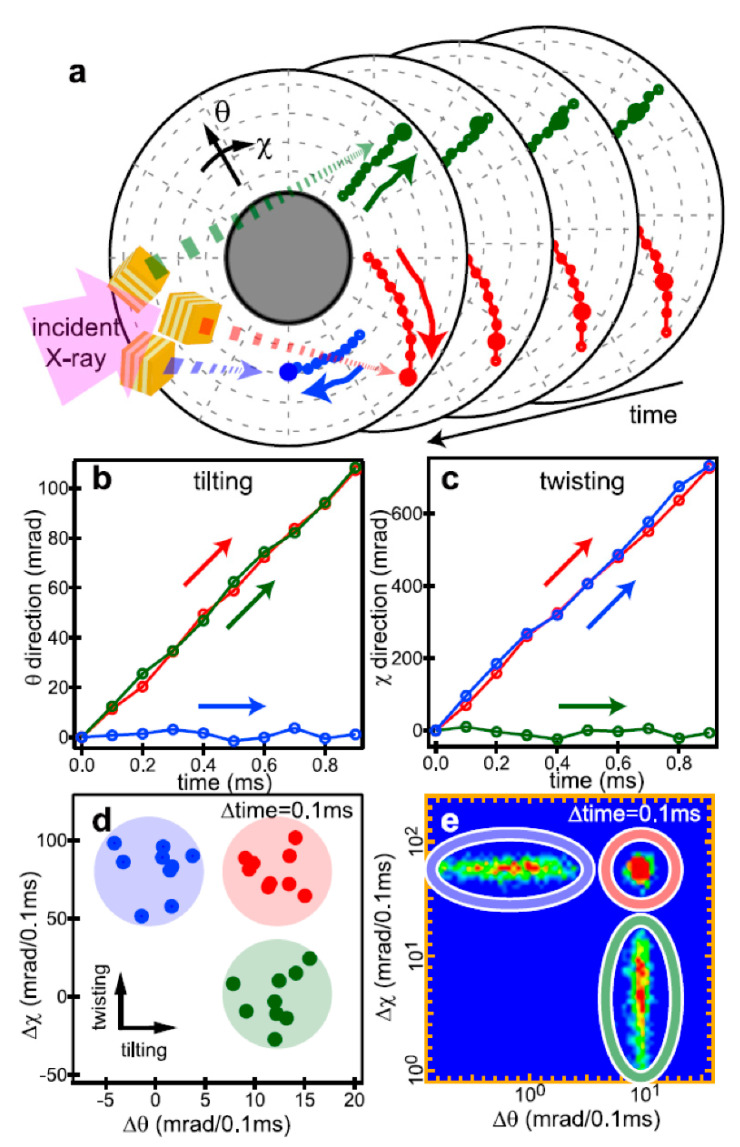
DXT analysis procedures range from diffraction spot trajectory to a 2D-internal motion map [26]. (**a**) Schematics of the diffraction spot trajectories in time-resolved diffraction images. Three examples of diffraction spot trajectories from gold nanocrystals exhibiting simple twisting motion (blue), simple tilting motion (green), and both tilting and twisting motions (red) are shown. (**b**,**c**) The displacement of angular position as a function of time in the tilting (**b**) and twisting (**c**) directions. (**d**) Two-dimensional scatterplot of the protein’s internal motion generated from angular displacement in both the tilting and twisting directions in a fixed time interval, e.g., 100 ms. (**e**) Two-dimensional-internal motion map of the protein. The logarithm of absolute displacement within a certain time interval was used to draw the 2D-histogram, which was normalized to obtain the 2D-internal motion map (probability density map).

**Figure 13 ijms-24-14829-f013:**
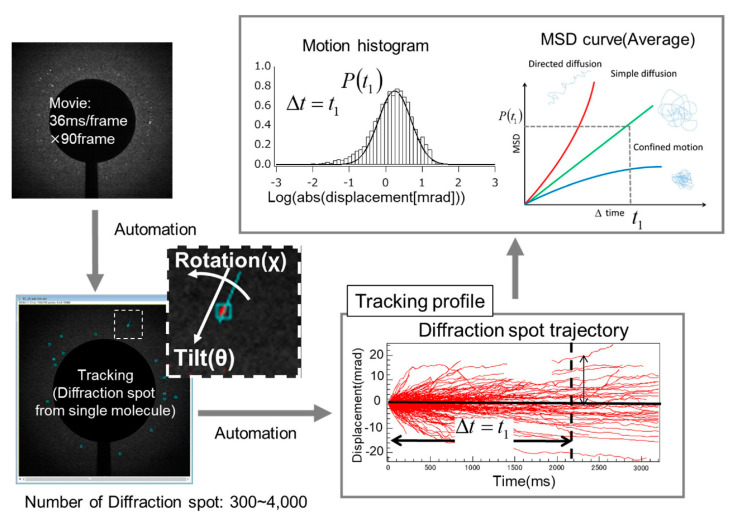
Data analysis strategy for DXT, where the diffraction point tracking is divided into the θ-axis and χ-axis along the time axis and the distribution of the diffraction point motions at a certain time point is calculated. For this purpose, 100–10,000 diffraction points provide credible data. The distribution is basically Gaussian.

**Figure 14 ijms-24-14829-f014:**
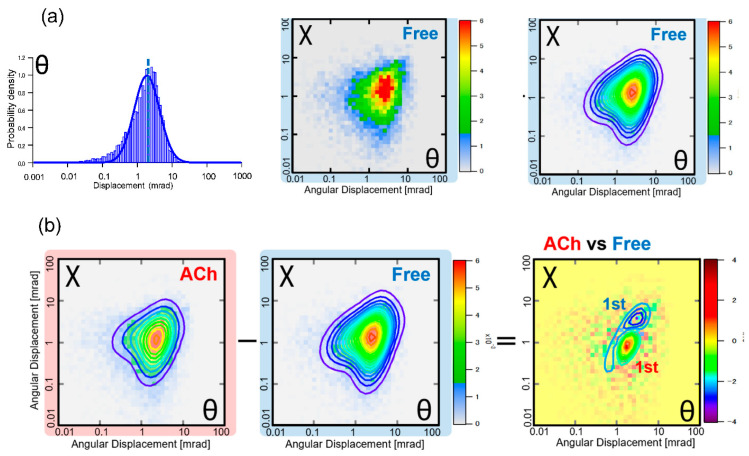
(**a**) One-dimensional motion distributions in the θ and χ directions are described at each time; however, the θ-χ two-dimensional motion distribution description is more accurate in monitoring the motion characteristics. (**b**) Two-dimensional motion distribution calculated for each aqueous condition; the 2D difference notation can be used to extract function-specific motion characteristics. Here, it is shown that there is a clear 2D motion difference between the presence and absence of ACh (ACh-free) molecules in an aqueous solution.

**Figure 15 ijms-24-14829-f015:**
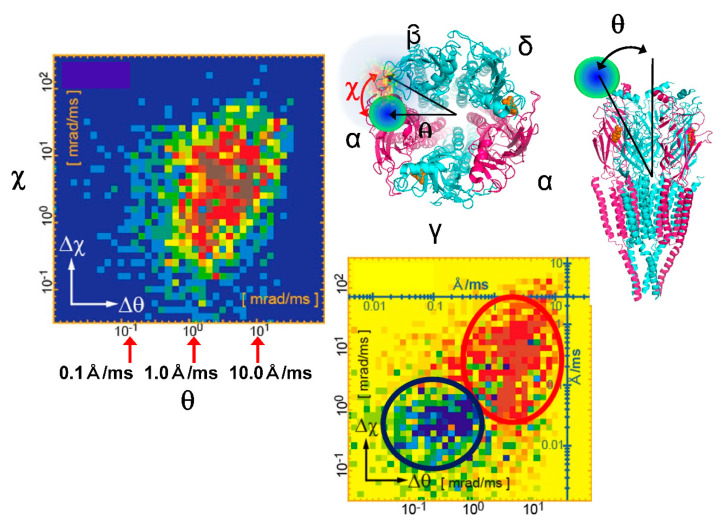
Two-dimensional dynamic distribution representation of the high sensitivity. Even picometer movements can be measured with high accuracy. Two-dimensional motion difference notations can clarify motion differences due to the function of the protein molecule.

**Figure 16 ijms-24-14829-f016:**
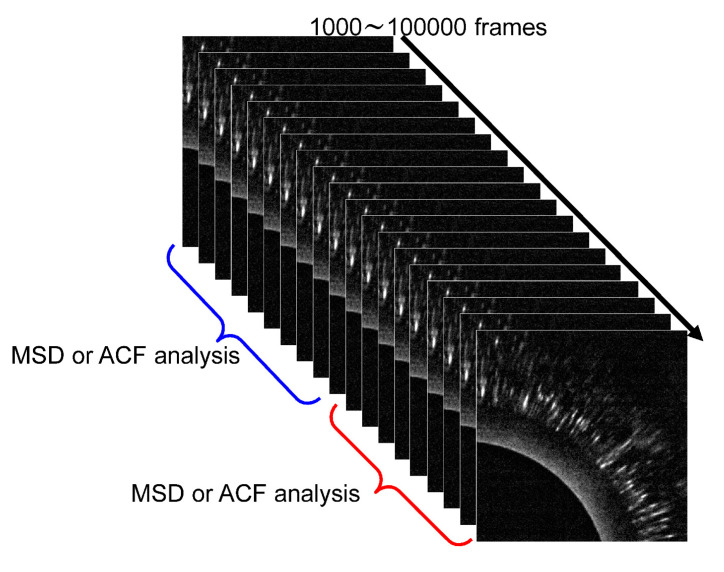
Continuous DXT typically measures 300–10,000 frames; the first half and the second half of the frames can be divided into MSD curve assessments to assess damage. When using synchrotron radiation, it is important to have a means of quantitatively assessing the damaging effects on the protein molecule. If damage is identified, the amount of synchrotron radiation flux needs to be reduced.

**Figure 17 ijms-24-14829-f017:**
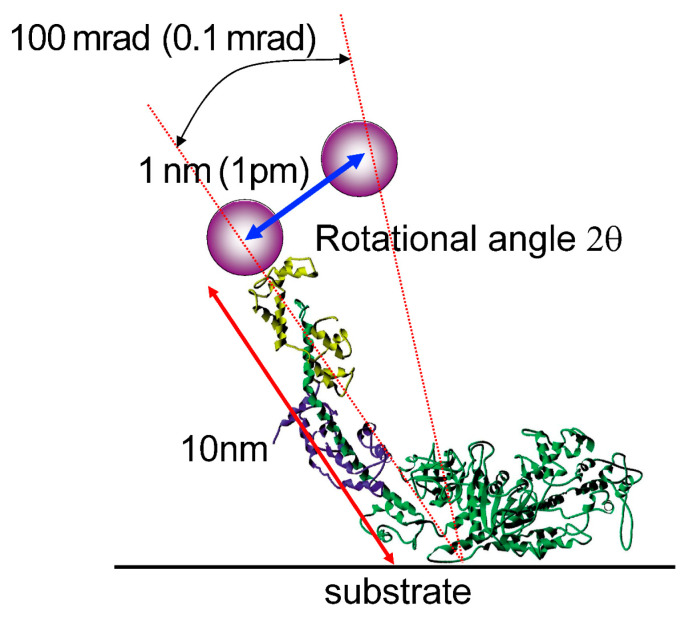
Conceptual diagram of the sensitive motion detection by DXT. The motion in the θ and χ directions is the basic rotational motion; when translational motion is detected by STM, AFM, etc., other noisy translational motion is removed by a vibration isolator, which is not perfect; DXT detects the rotational motion and does not require a special vibration isolator, which is a major advantage of the high sensitivity of DXT. When the length from the substrate to the end of the protein is 10 nm and the tip of the protein rotates by 100 mrad = 2θ, the translational equivalent is a lateral movement of 1 nm.

**Figure 18 ijms-24-14829-f018:**
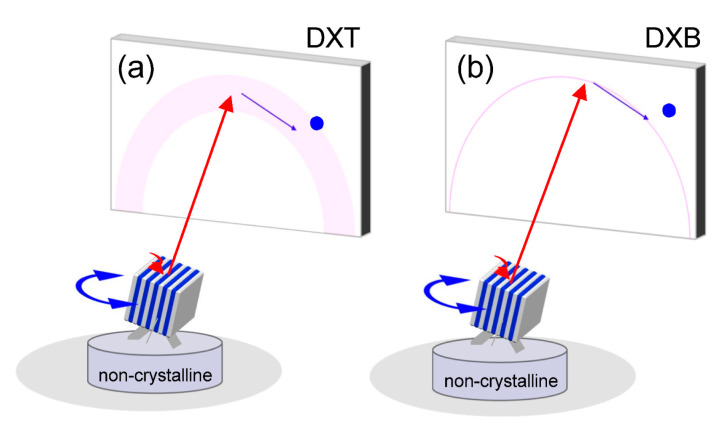
Extension from (**a**) DXT to (**b**) DXB (diffracted X-ray blinking). When viewing the motion of diffracted X-ray spots using monochromatic X-rays, the diffraction rings themselves appear to blink as the diffraction spots that deviate from the diffraction conditions disappear; this is called blinking. The extension from DXT to DXB enables molecular dynamics to be measured without synchrotron radiation and with a laboratory X-ray source.

**Figure 19 ijms-24-14829-f019:**
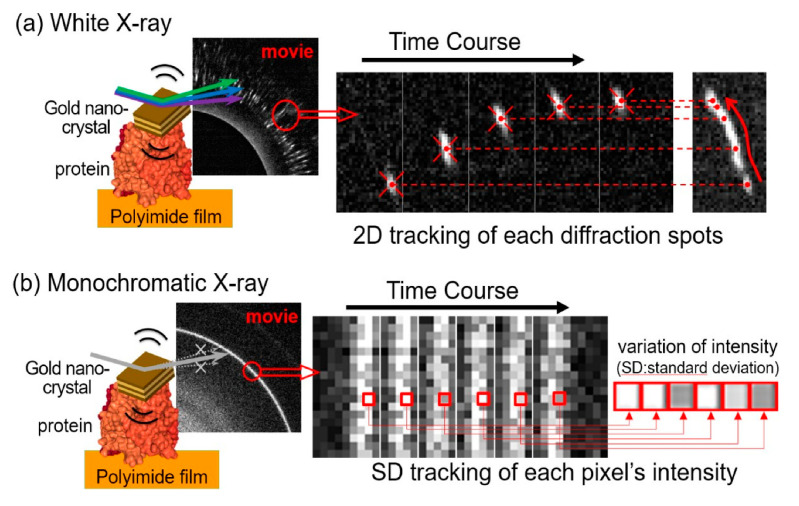
(**a**) The analysis principle of DXT is that only the position of diffraction spots on the 2D detector is followed; thus, the analysis is simple and virtually unaffected by X-ray intensity instability [42]. (**b**) Analysis principle of DXB, where the X-ray diffraction intensity variation within a single pixel is evaluated as an autocorrelation function [42]; however, care must be taken as the intensity instability at I zero directly affects the data.

**Figure 20 ijms-24-14829-f020:**
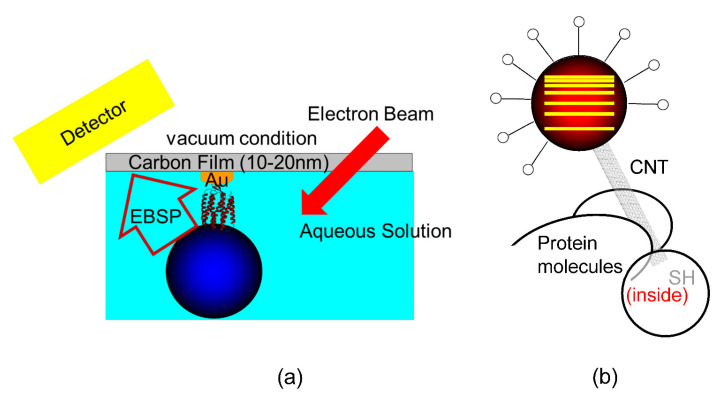
(**a**) DET (diffracted electron tracking), which is a successful development strategy of DXT/DXB; here, the X-ray is changed to an electron beam or even neutrons. (**b**) Fabrication of special nanocrystals that enable DXT to be measured even with monochromatic X-rays. This would enable the detection of continuous diffraction point shifts even in monochromatic X-rays.

## Data Availability

Not applicable.

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
