# Peer review of "Diffracted X-ray Tracking for Observing the Internal Motions of Individual Protein Molecules and Its Extended Methodologies"

_ijms, 2023, doi:10.3390/ijms241914829_

Round 1

Reviewer 1 Report

Comments and Suggestions for Authors

In this manuscript Sasaki reviews recent progress in the diffracted X-ray tracking (DXT) method. The author describes the method’s principle and demonstrates the superiority of the method by showing the author group’s studies on membrane proteins. This is an interesting work. The paper is well-written and clear for the readers to understand.

Therefore, I recommend the manuscript be published in International Journal of Molecular Sciences after revisions to address the following points, which would enhance readability.

Major points:

1. The author describes advantages of the DXT method over other methodologies. The temporal resolution can be found in line 58. However, it seems that θ and χ resolutions are not mentioned. To clearly show the advantages, please show their resolutions and the dynamic ranges as well as the temporal resolution on a table and compare them with those of other methodologies on the table.

2. The author describes the effects of nanocrystal labeling and X-ray irradiation on the DXT measurements in the manuscript. Concerning X-ray irradiation effect, an evaluation method is described in lines 282-290. I wonder how long a valid measurement time is in common. Also, I wonder whether x-ray irradiation to gold nanocrystals generates heat. If this is the case, to what extent does it damage proteins? Please mention these points in the manuscript.

Minor points:

1. “2. Labeled nanocrystals using DXT” (line 123) can be revised. It may be “Nanocrystal labeling for DXT.”

2. “DXT is a labeling method.” (line 124) can be revised. It may be “DXT requires a nanocrystal labeled to a target biomolecule.”

3. Some of the figures are adapted from publications. Please add statements of copyright permission to the figure legends if necessary.

Comments on the Quality of English Language

In my opinion, the English language is almost fine.

Author Response

Thank you for your excellent comments. I will respond to each one below:

Major point 1:

The resolution of θ and χ and dynamic range is not specific to the DXT method, and a general explanation is sufficient, so we decided to add the following text in the page 5-6:

The resolution of θ and χ, which are crucial for DXT measurements, is determined by the pixel size of the two-dimensional detector and the distance between the camera and the sample, known as the camera length. For instance [26], in our DXT measurements, we recorded time-resolved diffraction images using an X-ray image intensifier (V5445P, Hamamatsu Photonics) and a CMOS camera (1024 pixels x 1024 pixels, Photron FASTCAM SA1.1). This FASTCAM SA1.1 high-speed camera offers exceptional speed, capturing up to 9,000 frames per second, and delivers true 12-bit performance (dynamic range). The nominal entrance field of view for the X-ray image intensifier is 150 mm in diameter, with an effective pixel size of 0.1465 mm. With the incident X-ray's peak energy set at 15.2 keV and a sample-to-detector distance of 100 mm in our DXT setup, a one-pixel movement of a diffraction spot in the tilting (θ) direction corresponds to 0.7 mrad/pixel (@15.2 keV). For most diffraction spots originating from gold nanocrystals, situated 36.4 mm from the beam center, considering the d-spacing of Au (111) (d=2.35 Å), this distance corresponds to 248.5 pixels in our configuration. The circle with a radius of 248.5 pixels corresponds to approximately 1560 pixels in circumference. Consequently, a one-pixel shift in the twisting (χ) direction corresponds to 4.0 mrad/pixel @15.2 keV.

Major point2:

In numerous Differential X-ray Scattering (DXT) measurements, the phenomenon of gold nanocrystal heating induced by X-ray irradiation has not been substantiated. If localized temperature elevation in gold nanocrystals were to occur due to synchrotron radiation, it would manifest as an augmentation in the Brownian motion of these nanocrystals, and this augmented Brownian motion would be directly proportional to the duration of irradiation. However, in the multitude of DXT measurements conducted, such a unidirectional augmentation in motion has yet to be observed. This observation suggests that the absorption of X-rays by gold nanocrystals can be considered negligible. Added the following sentence to the text in the page 14:

Fortunately, given the absence of any observed unidirectional increase in the motion of gold nanocrystals in numerous DXT measurements conducted thus far, we posit that the consideration of thermal excitation of gold nanocrystals during synchrotron radiation is unnecessary.

Minor points:

We have made all the changes as you mentioned, and the copyright has been approved.

I would be happy to resubmit with the above corrections.

Reviewer 2 Report

Comments and Suggestions for Authors

This review article by Sasaki summarized the diffracted X-ray tracking for observing the internal motions of individual protein molecules and its extended methodologies.

Single-molecule measurements using visible light started in the 1970s and advanced the understanding of protein molecule movements in living cells. Fluorescence resonance energy transfer (FRET) was used for intramolecular protein motion monitoring but had limitations in measuring internal conformation changes. The need for atomic-level precision in observing protein molecule dynamics led to expectations for X-rays, electron beams, and neutrons.

Diffracted X-ray Tracking (DXT), proposed in 1998, involves labeling gold nanocrystals at sites of structural change and tracking diffraction spots over time. DXT allows the observation of real-space rotational motions of protein domains, providing dynamic information. It distinguishes between different models of internal molecular dynamics, such as tilting and twisting, by analyzing θ and χ motions. Gold nanocrystals are labeled at specific sites on protein molecules to facilitate tracking. The size of gold nanocrystals affects measurement accuracy, with smaller nanocrystals being preferred. Techniques for dispersing gold nanocrystals in aqueous solutions and modifying their surfaces for stability are crucial. Proteins of interest are fixed to substrates for DXT, enabling observation. Different methods, such as using antibodies or tags, help control protein orientation and immobilization. Substrate immobilization techniques are adaptable for observing membrane proteins in living cells. DXT data is processed statistically, with Brownian motion being a key phenomenon. Mean Square Displacement (MSD) curves are used to analyze motion modes, motion distribution, and ligand effects. The 2D motion information can be transformed into 3D structure change notations for functional analysis. DXT requires white X-rays with a specific energy range, limiting its accessibility. Diffracted X-ray Blinking (DXB) uses monochromatic X-rays to overcome this limitation, providing kinetic information. The analysis of DXB data is more complex due to X-ray intensity fluctuations. Efforts are underway to develop techniques like Small-Angle X-ray Scattering (SAXS) and Transmission X-ray Blinking (TXB) to address DXT limitations.

The author proposed the future developments as follows:

In vivo DXT/DXB is expected to contribute significantly to understanding biological phenomena.

Potential advances include adaptation to electron and neutron beams and the fabrication of special nanoparticles for monochromatic X-ray DXT.

The author summarized the historical development, principles, methods, and challenges associated with single-molecule dynamics using X-rays, with a focus on Diffracted X-ray Tracking (DXT) and its potential applications and advancements. In my opinion, the review article is well-organized. I recommend publication in its current form.

Author Response

Thanks for the great comments. I will repost the revised version to the other referee.
